# LOGIC PRE-TRAINING OF LANGUAGE MODELS

## ABSTRACT

Pre-trained language models (PrLMs) have been shown useful for enhancing a broad range of natural language understanding (NLU) tasks. However, the capacity for capturing logic relations in challenging NLU still remains a bottleneck even for state-of-the-art PrLM enhancement, which greatly stalled their reasoning abilities. Thus we propose logic pre-training of language models, leading to the logic reasoning ability equipped PrLM, PROPHET. To let logic pre-training perform on a clear, accurate, and generalized knowledge basis, we introduce *fact* instead of the plain language unit in previous PrLMs. The *fact* is extracted through syntactic parsing in avoidance of unnecessary complex knowledge injection. Meanwhile, it enables training logic-aware models to be conducted on a more general language text. To explicitly guide the PrLM to capture logic relations, three pre-training objectives are introduced: 1) logical connectives masking to capture sentence-level logics, 2) logical structure completion to accurately capture facts from the original context, 3) logical path prediction on a logical graph to uncover global logic relationships among facts. We evaluate our model on a broad range of NLP and NLU tasks, including natural language inference, relation extraction, and machine reading comprehension with logical reasoning. Results show that the extracted fact and the newly introduced pre-training tasks can help PROPHET achieve significant performance in all the downstream tasks, especially in logic reasoning related tasks.

## 1 INTRODUCTION

Machine reasoning in natural language understanding (NLU) aims to teach machines to understand human languages by building and analyzing the connections between the facts, events, and observations using logical analysis techniques like deduction and induction, which is one of the ultimate goals towards human-parity intelligence. Although pre-trained language models (PrLMs), such as BERT (Devlin et al., 2018), GPT (Radford et al., 2018), XLNet (Yang et al., 2019) and RoBERTa (Liu et al., 2019), have established state-of-the-art performance on various aspects in NLU, they are still short in complex language understanding tasks that involve reasoning (Helwe et al., 2021). The major reason behind this is that they are insufficiently capable of capturing logic relations such as negation (Kassner & Schütze, 2019), factual knowledge (Poerner et al., 2019), events (Rogers et al., 2020), and so on. Many previous studies (Sun et al., 2021; Xiong et al., 2019; Wang et al., 2020) are then motivated to inject knowledge into pre-trained models like BERT and RoBERTa. However, they too much rely on massive external knowledge sources and ignore that language itself is a natural knowledge carrier as the basis of acquiring logic reasoning ability (Ouyang et al., 2021). Taking the context in Figure 1 as an example, previous approaches tend to focus on entities such as the definition of "government" and the concepts related to it like "governor", but overlook the exact relations inherent in this example, thus failing to model the complex reasoning process.

Given the fact that PrLMs are the key supporting components in natural language understanding, in this work, we propose a fundamental solution by empowering the PrLMs with the capacity of capturing logic relations, which is necessary for logical reasoning. However, logical reasoning can only be implemented on the basis of clear, accurate, and generalized knowledge. Therefore, we leverage *fact* as the conceptual knowledge unit to serve the basis for logic relation extraction. *Fact* is organized as a triplet, i.e., in the form of predicate-argument structures, to represent the meaning such as "who-did-what-to-whom" and "who-is-what". Compared with existing studies that inject complex knowledge like knowledge graphs, the knowledge structure based on *fact* is far less complicated and more general in representing events and relations in languages.

On top of the fact-based knowledge structure, we present PROPHET, a logic-aware pre-trained language model to learn the logic-aware relations in a universal way from very large texts. In detail, we introduce three novel pre-training objectives based on the newly introduced knowledge structure basis *fact*: 1) logical connectives masking for learning sentence-level logic connection. 2) logical structure completion task on top of facts for regularization, aligning extracted fact with the original context. 3) logical path prediction to capture the logic relationship between facts. PROPHET is evaluated on a broad range of language understanding tasks: natural language inference, semantic similarity, machine reading comprehension, etc. Experimental results show that the fact is useful as the carrier for knowledge modeling, and the newly introduced pre-training tasks can improve PROPHET and achieves significant performance on downstream tasks.[1]

## 2   RELATED WORK

### 2.1   PRE-TRAINED LANGUAGE MODELS IN NLP

Large pre-trained language models (Devlin et al., 2018; Liu et al., 2019; Radford et al., 2018) have brought dramatic empirical improvements on almost every NLP task in the past few years. A classical norm of pre-training is to train neural models on a large corpus with self-supervised pre-training objectives. "Self-supervised" means that the supervision provided in the training process is automatically generated from the raw text instead of manually generation. Designing effective criteria for language modeling is one of the major topics in training pre-trained models, which decides how the model captures the knowledge from large-scale unlabeled data. The most popular pre-training objective used today is masked language modeling (MLM), initially used in BERT (Devlin et al., 2018), which randomly masks out tokens, and the model is asked to uncover it given surrounding context. Recent studies have investigated diverse variants of denoising strategies (Raffel et al., 2020; Lewis et al., 2020), model architecture (Yang et al., 2019), and auxiliary objectives (Lan et al., 2019; Joshi et al., 2020) to improve the model strength during pre-training. Although the existing techniques have shown effectiveness in capturing syntactic and semantic information after large-scale pre-training, they perform sensitivity to role reversal and struggles with pragmatic inference and role-based event knowledge (Rogers et al., 2020), which are critical to the ultimate goal of complex reasoning that requires to uncover logical structures. However, it is difficult for pre-trained language models to capture the logical structure inherent in the texts since logical supervision is rarely available during pre-training. Therefore, we are motivated to explicitly guide the model to capture such clues via our newly introduced self-supervised tasks.

### 2.2   REASONING ABILITY FOR PRE-TRAINED LANGUAGE MODELS

There is a lot of work in the research line of enhancing reasoning abilities in pre-trained language models via injecting knowledge. The existing approaches mainly design novel pre-training objectives and leverage abundant knowledge sources such as WordNet (Miller, 1995).

Notably, ERNIE 3.0 (Sun et al., 2021) uses a broad range of pre-training objectives from word-aware, structure-aware to knowledge-aware tasks, based on a 4TB corpus consisting of plain texts and a large-scale knowledge graph. WKLM (Xiong et al., 2019) replaces entity mentions in the document with other entities of the same type, and the objective is to distinguish the replaced entity from the original ones. KEPLER (Wang et al., 2021b) encodes textual entity descriptions using embeddings from a PrLM to take full advantage of the abundant textual information. K-Adapter (Wang et al., 2020) designs neural adapters to distinguish the type of knowledge sources to capture various knowledge.

Our proposed method differs from previous studies in three aspects. Firstly, our model does not require any external knowledge resources like previous methods that use WordNet, WikiData, etc. We only use small-scale textual sources following the standard PrLMs like BERT (Devlin et al., 2018), along with an off-the-shelf dependency parser to extract facts. Secondly, previous works only consider triplet-level pre-training objectives. We proposed a multi-granularity pre-training strategy, considering not only triplet-level information but also sentence-level and global knowledge to enhance logic reasoning. Finally, we propose a new training mechanism apart from masked language modeling (MLM), hoping to shed light on more logic pre-training strategies in this research line.

---

[1]Our codes have been uploaded as supplemental material, which will be open after the double review period.

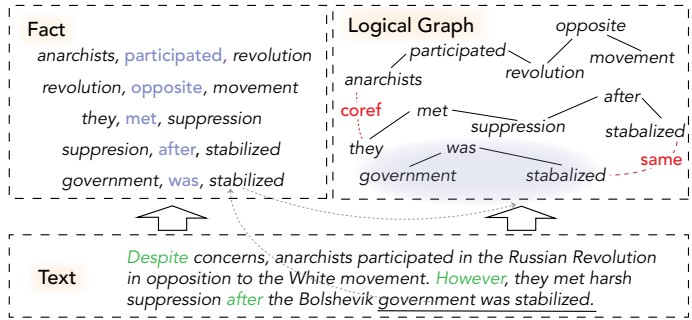

Figure 1: How the facts and logical graph constructed from raw text inputs. Edges in red denotes additional edges added in the logical graph, while text with green indicates the sentence-level logical connectives which will be mentioned in §4.

## 3 PRELIMINARIES

In this section, we will introduce the concept of *fact* and *logical graph*, which is the basis of PROPHET. We will also describe extracting the fact for logical graph construction, as an example shown in Figure 1.

### 3.1 FACT

Following Nakashole & Mitchell (2014) and Ouyang et al. (2021), we extract facts which are triplets represented as $T = \{A_1, P, A_2\}$, where $A_1$ and $A_2$ are the arguments and $P$ is the predicate between them. It can well represent a broad range of *facts*, reflecting the notion of "who-did-what-to-whom" and "who-is-what", etc.

We extract such facts in a syntactic way, which makes our approach generic and easy to apply. Given a document, we first split the document into multiple sentences. For each sentence, we conduct dependency parsing using StanfordCoreNLP (Manning et al., 2014).[2] For the analyzed dependencies, basically, we consider verb phrases and some prepositions in the sentences as "predicates", and then we search for their corresponding actors and actees as the "arguments".

### 3.2 LOGICAL GRAPH

A logical graph is an undirected (but is not required to be connected) graph that represents logical dependency relation between components in facts. In logical graphs, nodes represent argument/predicates in the fact, and edges indicate whether two nodes have relations in a fact. Such a structure can well unveil and organize semantic information captured by facts. Besides, a logical graph supports considerations among long-range dependencies via connecting arguments and their relations in different facts across different spans.

We further show how to construct such graphs based on facts. Despite given relations in facts, we design another two types of edges based on identical mentions and coreference information. (1) There can be identical mentions in different sentences, resulting in repeated nodes in facts. We connect nodes corresponding to the same non-pronoun arguments by edges with edge type *same*. (2) We conduct coreference resolution on context using an off-to-shelf model to identify arguments in facts that refer to the same one.[3] We add edges with type *coref* between them. The final logical graph is denoted as $S = (V, E)$, where $V = A_i \cup P$ and $i \in \{1, 2\}$.

---

[2]https://stanfordnlp.github.io/CoreNLP/, we also tried to use OpenIE directly; however, the performance is not satisfactory.

[3]https://github.com/huggingface/neuralcoref.

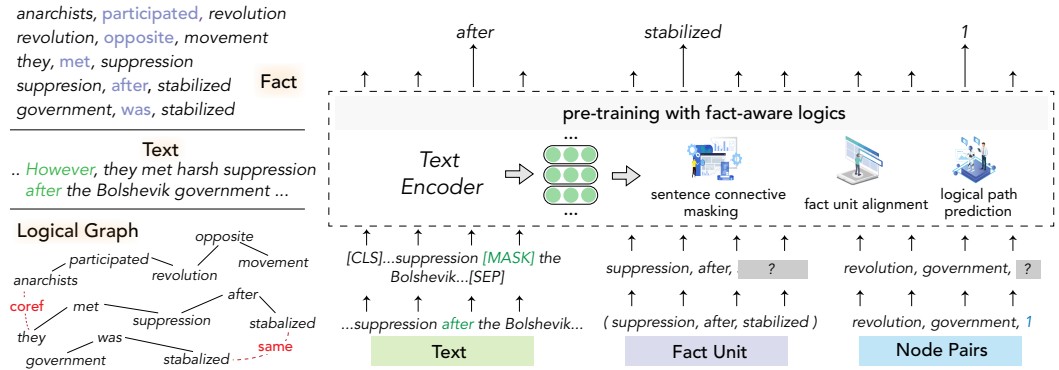

Figure 2: An illustration about pre-training methods used in PROPHET. The model takes the text, extracted *fact* and the randomly sampled node pairs in the logical graph as the input. The model is pre-trained with three novel objectives. One is the standard masked language modeling applied to sententious connectives, the others are fact alignment and logical path prediction.

# 4 PROPHET

## 4.1 MODEL ARCHITECTURE

We follow BERT (Devlin et al., 2018) and use a multi-layer bidirectional Transformer (Vaswani et al., 2017) as the model architecture of PROPHET. For keeping the focus on the newly introduced techniques, we will not review the ubiquitous Transformer architecture in detail. We develop PROPHET by using exactly the same model architecture as BERT-base, where the model consists of 12 transformer layers, with 768 hidden size, 12 attention heads, and 110M model parameters in total.

## 4.2 LOGIC-AWARE PRE-TRAINING TASKS

We describe three pre-training tasks used for pre-training PROPHET in this section. Figure 2 is an illustration of PROPHET pre-training. The first task is logical connectives masking (LCM) generalized from masked language modeling (Devlin et al., 2018) for logical connectives to learn sentence-level representation. The second task is logical structure completion (LSC) for learning logic relationship inside a fact, where we first randomly mask arguments in facts, and then predict those items. Finally, a logical path prediction (LPP) task is proposed for recognizing the logical relations of randomly selected node pairs.

**Logical Connective Masking** Logical connective masking is an extension of the masked language modeling (MLM) pre-training objective in Devlin et al. (2018), with a particular focus on connective indication tokens. We use the Penn Discourse TreeBank 2.0 (PDTB) (Prasad et al., 2008) to draw the logical relations among sentences. Specifically, PDTB 2.0 contains relations that are manually annotated on the 1 million Wall Street Journal (WSJ) corpus and are broadly characterized into "Explicit" and "Implicit" connectives. We use the "Explicit" type (in total 100 such connectives), which apparently presents in sentences such as discourse adverbial "*instead*" or subordinating conjunction "*because*". Taking all the identified connectives and some randomly sampled other tokens (for a total 15% of the tokens of the original context), we replace them with a *[MASK]* token 80% of the time, with a random token 10% of the time and leave them unchanged 10% of the time. The MLM objective is to predict the original tokens of these sampled tokens, which has proven effective in previous works (Devlin et al., 2018; Liu et al., 2019). In this way, the model learns to recover the logical relations for two given sentences, which helps language understanding. The objective of this task is denoted as $\mathcal{L}_{conn}$.

**Logical Structure Completion** To align representation between the context and the extracted fact, we introduce a pre-training task of logical structure completion. The motivation here is to encourage

the model to learn the structure-aware representation that encodes the "Who-did-What-to-Whom"-like meanings for better language understanding. To speak in detail, we randomly select a specific proportion $\lambda$ of the total facts ($\lambda = 20\%$ in this work), from a given context. For each chosen fact, we either ask the model to complete "*Argument-Predicate-?*" or "*Argument-?-Argument*" (the templates are selected based on equal probability). We denote all the blanks that need to be completed as $m^a$ and $m^p$, denoting arguments and predicates, respectively. In our implementation, this objective is the same as masked language modeling to keep simplicity, by using the original loss following Devlin et al. (2018).

$$\mathcal{L}_{align} = -\sum_{i \in a \cup p} \log D(x_i | m^a, m^p), \tag{1}$$

where $D$ is the discriminator to predicts a token from a large vocabulary.

**Logical Path Prediction**   To learn representation from the constructed logical graph, thus endowing the model with global logical reasoning ability, we propose the pre-training task of predicting whether there exists a path between two selected nodes in the logical graph. In this way, the model learns to look at logical relations across a long distance of arguments and predicates in different facts.

We randomly sample 20% nodes from logical graph to form set $V'$, there are in total $C^2_{|v'|}$ node pairs. We have a maximum number $max_p$ of node pairs to predict. To avoid bias in the training process, we try to make sure that $\frac{max_p}{2}$ are positive samples and the rest are negative samples, thus balancing positive-negative ratios. If the number of positive/negative samples is less than $\frac{max_p}{2}$, we just keep the original pairs. Formally, the pre-training objective of this task is calculated as below following Guo et al. (2020):

$$\mathcal{L}_{Path} = -\sum_{v_{i,j} \in V'} [\delta \log \sigma[v_i, v_j] + (1 - \delta) \log(1 - \sigma[v_i, v_j])], \tag{2}$$

where $\delta$ is 1 when $v_i$ and $v_j$ have a connected path and 0 otherwise. $[v_i, v_j]$ denotes the concatenation of representations of $v_i$ and $v_j$.

The final training objective is the weighted sum of the above mentioned three losses.

$$\mathcal{L} = \mathcal{L}_{conn} + \mathcal{L}_{align} + \mathcal{L}_{Path}. \tag{3}$$

### 4.3 PRE-TRAINING DETAILS

We use the English Wikipedia (1.1 million articles in total), we sample the train and valid datasets with a split ratio of 19 : 1 on the original datasets. We omit the "Reference" and "Literature" part in a document to ensure data quality. Following the previous practice (Devlin et al., 2018), we limit the length of sentences in each batch as up to 512 tokens and the batch size is 128. We use Adam (Kingma & Ba, 2014) with $\beta_1 = 0.9$, $\beta_2 = 0.98$ and $\epsilon = 1e-6$, and weight decay is set as 0.01. We pre-train our model for $500k$ steps. We use 8 NVIDIA V100 32G GPUs, with FP16 and deepspeed for training acceleration. Initialized by the pre-trained weights of BERT$_{base}$, we continue training our models for $200k$ steps.

## 5 EXPERIMENTS

### 5.1 TASKS AND DATASETS

Our experiments are conducted on a broad range of language understanding tasks, including natural language inference, machine reading comprehension, semantic similarity, and text classification. Some of these tasks are a part of GLUE (Wang et al., 2018) benchmark. We also extend our experiments to DocRED (Yao et al., 2019), a widely used benchmark of document-level relation extraction for generalizability. To verify our model's reasoning abilities of logic, we perform experiments on two recent logical reasoning datasets in the form of machine reading comprehension, ReClor (Yu et al., 2020) and LogiQA (Liu et al., 2020).

| Model | Classification | | Language Inference | | | Semantic Similarity | | | Avg. |
|---|---|---|---|---|---|---|---|---|---|
| | CoLA | SST-2 | MNLI | QNLI | RTE | MRPC | QQP | STS-B | - |
| *In literature* | | | | | | | | | |
| BERT$_{base}$ | 52.1 | 93.5 | 84.6/83.4 | 90.5 | 66.4 | 88.9 | 71.2 | 85.8 | 79.6 |
| SemBERT$_{base}$ | 57.8 | 93.5 | 84.4/84.0 | 90.9 | 69.3 | 88.2 | 71.8 | 87.3 | 80.8 |
| *Our implementation* | | | | | | | | | |
| BERT$_{base}$ | 53.6 | 93.5 | 84.6/83.4 | 90.9 | 66.6 | 88.6 | 71.2 | 85.8 | 79.8 |
| PROPHET | 57.0 | 93.9 | 85.3/84.3 | 91.4 | 69.8 | 89.5 | 72.0 | 86.0 | 81.1 |

Table 1: Leaderboard results on GLUE benchmark. The number below each task denotes the number of training examples. F1 scores are reported for QQP and MRPC, Spearman correlations are reported for STS-B, and accuracy scores are reported for the other tasks.

| Model | ReClor | | | | LogiQA | |
|---|---|---|---|---|---|---|
| | Dev | Test | Test-E | Test-H | Dev | Test |
| Human Performance* | - | 63.0 | 57.1 | 67.2 | - | 86.0 |
| *In literature* | | | | | | |
| FOCAL REASONER (Ouyang et al., 2021) | 78.6 | 73.3 | 86.4 | 63.0 | 47.3 | 45.8 |
| LReasoner (Wang et al., 2021a) | 74.6 | 71.8 | 83.4 | 62.7 | 45.8 | 43.3 |
| DAGN (Huang et al., 2021) | 65.8 | 58.3 | 75.9 | 44.5 | 36.9 | 39.3 |
| BERT$_{large}$ (Devlin et al., 2018) | 53.8 | 49.8 | 72.0 | 32.3 | 34.1 | 31.0 |
| XLNet$_{large}$ (Yang et al., 2019) | 62.0 | 56.0 | 75.7 | 40.5 | - | - |
| RoBERTa$_{large}$ (Liu et al., 2019) | 62.6 | 55.6 | 75.5 | 40.0 | 35.0 | 35.3 |
| DeBERTa$_{large}$ (He et al., 2020) | 74.4 | 68.9 | 83.4 | 57.5 | 44.4 | 41.5 |
| *Our implementation* | | | | | | |
| BERT$_{base}$ | 51.2 | 47.3 | 71.6 | 28.2 | 33.8 | 32.1 |
| PROPHET | 53.4 | 48.8 | 72.4 | 32.2 | 35.2 | 34.1 |

Table 2: Accuracy on ReClor and LogiQA dataset. The public methods are based on large models.

## 5.2 RESULTS

Table 1 shows results on the GLUE benchmark datasets. We have the following observations from the above results.

(1) PROPHET obtains substantial gains over the BERT baseline (continual trained for 200K steps for a fair comparison), indicating that our model can work well in a general sense of language understanding.

(2) PROPHET performs particularly well on language inference tasks including MNLI, QNLI, and RTE,[4] which indicates our model's ability to reasoning.

(3) Whether it is large-scale datasets such as QQP and MNLI or small datasets like COLA and SST-B, our model demonstrates a consistent improvement, indicating its robustness.

(4) From Table 2, we can see that PROPHET improves the logical reasoning ability of BERT baseline by a large margin. Especially, armed with our approach, the results on the two datasets for the BERT-base model are comparable or even surpass those with BERT-large results.

In addition, we conducted experiments on a large-scale human-annotated dataset for document-level relation extraction (Yao et al., 2019). The results are shown in Table 3.[5] From the table, we can see that PROPHET still does well for relation extraction for documents by outperforming the baseline

---

[4]We exclude the problematic WNLI set.

[5]We only report the results for Ign F1 in the annotated setting as the distant supervision is too slow to train.

| Model | Dev | | Test | |
|---|---|---|---|---|
| | F1 | Intra-F1 | Inter-F1 | F1 |
| BERT$_{base}$* (Devlin et al., 2018) | 54.2 | 61.6 | 47.2 | 53.2 |
| Two-Phase BERT* (Wang et al., 2019) | 54.4 | 61.8 | 47.3 | 53.9 |
| PROPHET | 54.8 (↑0.6) | 62.4 (↑0.8) | 47.5 (↑0.3) | 54.3 (↑1.1) |

Table 3: Main results on the dev and test set for DocRED. * indicates that the results are taken from Nan et al. (2020). Intra- and Inter-F1 indicates F1 scores for the intra- and inter-sentence relations following the setting of Nan et al. (2020).

| Model | Classification | | Language Inference | | | Semantic Similarity | | | Avg. |
|---|---|---|---|---|---|---|---|---|---|
| | CoLA | SST-2 | MNLI | QNLI | RTE | MRPC | QQP | STS-B | - |
| PROPHET | 57.0 | 93.9 | 85.3/84.3 | 91.4 | 69.8 | 89.5 | 72.0 | 86.0 | 81.1 |
| w/o LCM | 53.6 | 93.5 | 85.1/84.0 | 90.9 | 68.2 | 88.6 | 71.2 | 85.8 | 80.1 |
| w/o LSC | 53.6 | 93.6 | 85.0/84.1 | 91.3 | 69.0 | 88.9 | 71.4 | 85.9 | 80.3 |
| w/o LPP | 52.1 | 93.0 | 84.6/83.4 | 90.9 | 66.4 | 88.6 | 71.2 | 85.8 | 79.6 |

Table 4: Ablation studies of PROPHET on the test set of GLUE dataset.

substantially. It even surpasses the two-phase BERT. Also, our model is especially good at coping with inter-sentence relations compared with baseline models, which means that our model is indeed capable of synthesizing the information across multiple sentences of a document, verifying the effectiveness of leveraging sententious and global information.

# 6 ANALYSIS

## 6.1 ABLATION STUDY

To investigate the impacts of different objectives introduced, we evaluate three variants of PROPHET as described in Section 4.2: 1) the **w/o LCM** model adopts a substitute without logical connectives masking as the pre-training objective, 2) the **w/o LSC** model is such that it leaves out the logical structure completion objective, and 3) the **w/o LPP** model only uses the objectives of connective masking and structure completion. The results are shown in Table 4.

Based on the ablation studies, we come to the following conclusions. Firstly, all three components contribute to the performance as removing any one of them causes a performance drop on the average score. Especially, the average point drops the most as we remove the logical path prediction objective, which sheds light on the importance of modeling chain-like relations of events. Secondly, we can see that logical path prediction contributes the most to the reasoning abilities as the performance on language inference improves the most when we add the sententious connective masking objective and the task of logical path prediction.

## 6.2 COMPARISON BETWEEN FACT AND ENTITY-LIKE KNOWLEDGE

We also replace the injected fact with common practice using entity-like knowledge, which is using named entities. In detail, we change the arguments in facts into named entities recognized by StanfordCoreNLP,[6] and leave the predicates extracted unchanged, resulting in the form of $< NE_1, predicate, NE_2 >$ (NE stands for named entity). If a fact is not recognized with any named entities, we just leave it out.

The results are shown in Table 5. We can see that the performance is hurt a lot, even worse than vanilla BERT. This is quite intuitive as the number of named entities is far less than our obtained fact,

---
[6]https://stanfordnlp.github.io/CoreNLP/

| Model | Classification | | Language Inference | | | Semantic Similarity | | | Avg. |
|---|---|---|---|---|---|---|---|---|---|
| | CoLA | SST-2 | MNLI | QNLI | RTE | MRPC | QQP | STS-B | - |
| PROPHET | 57.0 | 93.6 | 85.0/84.1 | 91.4 | 69.8 | 89.2 | 71.4 | 86.0 | 81.0 |
| w/ named entities | 50.4 | 93.2 | 84.9/84.2 | 90.8 | 68.7 | 88.4 | 71.0 | 84.9 | 79.3 |

Table 5: Results on GLUE test set when replacing facts with named entities and key the relations unchanged.

missing a lot of information inherent in the context. Whereas our introduced fact can well capture the knowledge used in the reasoning process, providing a fundamental reasoning basis.

### 6.3 ATTENTION MATRIX HEATMAP

We plot the attention matrix in token level to see how our model interprets the context using heatmap shown in Figure 3.

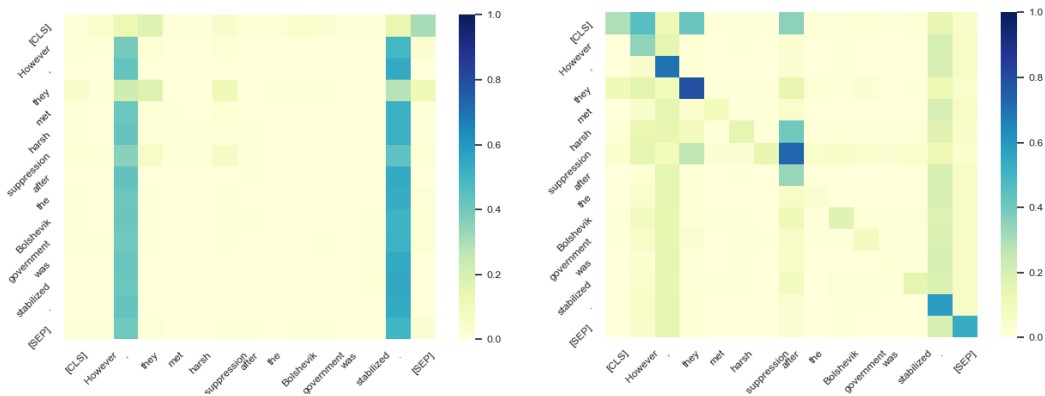

Figure 3: Heatmap of the attention matrix of vanilla BERT and our implemented PROPHET for the sentence "However, they met harsh suppression after the Bolthevik government was stabilized.". Weights are selected from the first head of the last attention layer.

From the figure, we can see that the vanilla BERT attends to delimiters, particularly punctuation as suggested in Clark et al. (2019). In comparison, our model exhibits quite different attention distribution. Firstly, our model clearly decreases the influences introduced by punctuation. Secondly, our model pays more attention to tokens representing discourse-level information, such as "however" and "after", which is consistent with our motivation. It also well captures the relations of pronouns. The event characteristics are also illustrated as seen from the "after suppression" phrases.

### 6.4 EFFECT OF DIFFERENT CONTEXT LENGTH

We group samples into ten subsets according to an equal amount of samples (around 1000 samples per interval) by context length since the majority of the samples concentrate on the interval of under 60. The statistics of MNLI-matched and MNLI-mismatched dev sets are shown in Table 6. Then we calculate the accuracy of the baseline and PROPHET per group for both the matched and mismatched set, as shown in Figure. 4. We observe that the performance of the baseline groups drops dramatically when encountered with long contexts, especially for those longer than 45 tokens, while our model performs more robustly on those intervals (the slope of the dashed line is more gentle).

### 6.5 CASE STUDY

We also give a case study to demonstrate that PROPHET could enhance the reasoning process in language understanding. Given two sentences, we use PROPHET and BERT-base to predict whether

Table 6: Distribution of context length on dev set of MNLI-matched and MNLI- mismatched dataset.

| Dataset | [0, 29) | [30, 59) | [60, 89) | [90, 119) | [120, 149) | [150, 179) | [180, 209) | [210, 239) |
|---|---|---|---|---|---|---|---|---|
| MNLI-matched | 39.2% | 49.8% | 9.6% | 1.0% | 0.12% | 0.12% | 0.02% | 0.06% |
| MNLI-mismatched | 33.7% | 55.0% | 9.7% | 1.7% | 0.3% | 0.1% | 0.1% | 0% |

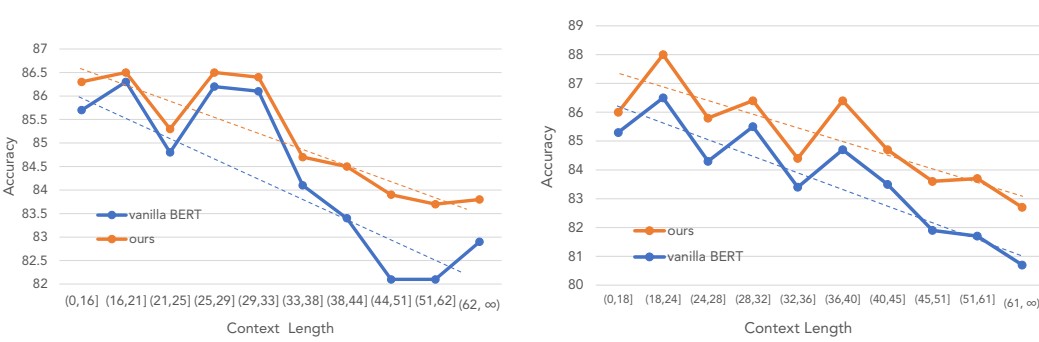

Figure 4: Accuracy of different context length on MNLI-match (left) and MNLI-mismatch (right) dev set. There are approximate 1000 samples in each intervals.

the sentences are entailed or not. Results are shown in Figure 5. To see the language understanding ability of our model, we made two subtle changes in the original training sample. Firstly, we change the entity referred to in the sentence. We can see that PROPHET learns better alignment relations between entities than BERT-base model. Additionally, we add a negation in the sentence. Although this change is small, it completely changes the semantic of the sentence, and leads to a reversal of the ground-truth labels. We can see that PROPHET is good at all the samples given, indicating that it is not only good at reasoning in language understanding, but also is more robust than baseline models.

| | *Unchanged* | *Entity change* | *Negation* |
|---|---|---|---|
| Input | Sentence 1: *Note that SBB, CFF and FFS stand out for the main railway company, in German, French and Italian..*

Sentence 2: *The French railway company is called SNCF..* | Sentence 1: *Note that SBB, CFF and FFS stand out for the main railway company, in German, French and Italian..*

Sentence 2: *The French railway company is called SBB..* | Sentence 1: *Note that SBB, CFF and FFS stand out for the main railway company, in German, French and Italian..*

Sentence 2: *The French railway company is not called SNCF..* |
| Label | not entailment | not entailment | entailment |
| Prediction | Prophet: not entailment ✓

BERT-base: not entailment ✓ | Prophet: not entailment ✓

BERT-base: entailment ✗ | Prophet: nentailment ✓

BERT-base: not entailment ✗ |

Figure 5: We take an example from RTE dataset, and use PROPHET and BERT-base to predict the label of the relations among two given sentences.

## 7 CONCLUSION

In this paper, we leverage *fact* in a newly pre-trained language model PROPHET to capture logic relations essentially, in consideration of the fundamental role of PrLM serving in NLP and NLU tasks. We introduce three novel pre-training tasks and show that PROPHET achieves significant improvement over various logic reasoning involved NLP and NLU downstream tasks, including language inference, sentence classification, semantic similarity, and machine reading comprehension. Further analysis shows that our model can well interpret the inner logical structure of the context to aid the reasoning process.

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
