# OpenReview forum: "Logic Pre-Training of Language Models"
_ICLR.cc/2022/Conference — ICLR 2022 Submitted_

### Official Review · Reviewer_K1pp · 2021-10-28

**Correctness:** 2
**Technical Novelty And Significance:** 1
**Empirical Novelty And Significance:** 2
**Recommendation:** 3
**Confidence:** 5

**Main Review:**

I think this paper has the following weak points:
1. The word "logic" is misused. Obviously, the paper considers the additional information of dependency parsing rather than logic. Secondly, the information in this paper is not "fact", they should be called word dependency. Of course, the naming is not a very serious issue. I just hope that the author's naming will not mislead the readers.
2. The main drawback of this paper is that it does not convince me that dependency parsing is a very important additional information for pre-training language models. According to previous studies [1], the pre-training of BERT actually already contains morphology, syntax, and semantics information. And the current state-of-the-art dependency parser is even built based on pre-trained language models. This is what I suspect motivates the further introduction of dependency information. In particular, this paper does not specify what exactly the introduction of dependency parsing information solves for the original masked language modeling task. For reference, [2] introduces span information (simpler morphology information than dependency), thus making pre-training more reasonable and difficult.
3. Equation (1) should be rewriten. Since ma and mp are blanks, why can they be used as conditions?
4. The logical connective masking proposed by the authors is very strange. It is irrelevant to their motivation based on fact and logical graph as stated in the previous sections.
5. I do not think that the authors' experiments support their motivation very well. First, with reference to SpanBERT, which introduces span information, the effect of PROPHET is degraded. Second, the authors do not experimentally verify what exactly the newly introduced dependency parsing solves. The example in Figure 5 does not give an in-depth insight to me.
6. In the experiments of Figure 3, it is not reasonable to compare the attention matrix of token level only. Because some dependency information may be represented by higher levels of BERT.
7. in equation 3, the authors did not consider the original masked language modeling as the loss, which I think should be helpful.


[1] Bert rediscovers the classical nlp pipeline
[2] SpanBERT: Improving Pre-training by Representing and Predicting Spans

**Summary Of The Paper:**

This paper proposes to add the dependency parsing information of sentences as part of the pre-training objectives to the pre-training of BERT. In particular, it designs three pre-training tasks: logical connective masking, logical structure completion, and logical path prediction, to introduce the dependency parsing information.

**Summary Of The Review:**

In summary, I think the motivation for adding additional dependency parsing has not been adequately articulated. And the experiments have not fully verified what the dependency parsing information actually solves. The simpler SpanBERT is in fact better than this paper. Therefore, I think this paper does not reach the threshold of ICLR.

---

### Official Review · Reviewer_rHDy · 2021-10-31

**Correctness:** 3
**Technical Novelty And Significance:** 2
**Empirical Novelty And Significance:** 2
**Recommendation:** 5
**Confidence:** 4

**Main Review:**

This paper is intended to inject knowledge into pre-trained language models without relying on external knowledge. Instead, it proposes to leverage knowledge in the corpus itself via dependency parsing and pre-trains a new language model with 3 designed pre-training tasks.

The authors should be cautious when using big words such as “logic reasoning”. The writing in sections 1 and 3 gives the feeling of overclaiming. It does not specify the exact meaning of the word "logic". The related work section is far from being exhaustive, missing a large portion of PrTM with syntax information. The pretraining tasks lack some details. Is LSC another modified MLM? Does it consider context? This work is based on BERT-base, which has limited capacity. It is not convincing of where the improvement is from, and it is unlikely that the proposed base-size model will have a broader impact. In short, I find the overall performance of the method not impressive. Additionally, in the analysis section, the attention maps are clearly selected biased over the multi-heads. It is not difficult to select the desired heat map from multiple heads.

Strengths:
1. This paper proposes a new method to enhance the logical reasoning ability of PrTM via syntactic parsing that avoids complex knowledge injection.
2. Experiments show the effectiveness of the method, and the writing is clear and easy to understand.

Weakness:
1. The proposed method is not novel enough when compared with other works using syntactic parsing to enhance PrTMs.
2. It is likely that there are only very limited logic relations contained in texts, making PROPHET not substantially beneficial to downstream NLP tasks.
3. The experimental results are not convincing enough. Firstly, this paper only conducts experiments on BERT-base, which is an undertrained model. It is worth at least reporting both BERT-base and BERT-large to demonstrate the scalability of PROPHET.




**Summary Of The Paper:**

To enable pre-trained language models to capture logic relations contained in natural language, this paper presents a pre-trained language model, PROPHET, which aims to encode logic relations (facts) with three pre-training objectives. Experiments show that the proposed method outperforms its baseline BERT-base in a broad range of NLP tasks.

**Summary Of The Review:**

Injecting knowledge into pre-trained models without resorting to external knowledge sources is interesting, and the effect is confirmed by empirical results. However, my concern is the limited novelty of the proposed model.

---

### Official Review · Reviewer_DQM2 · 2021-11-02

**Correctness:** 2
**Technical Novelty And Significance:** 2
**Empirical Novelty And Significance:** 2
**Recommendation:** 3
**Confidence:** 5

**Main Review:**

## The good

The paper attempts to build logic-aware models by pre-training on relation prediction and link prediction objectives. Since natural language corpuses are not annotated with facts (triples of source, sink and relation), the paper proposes a dependency parser based extraction mechanism. The paper investigates their method on GLUE benchmark, and also investigates it on two additional datasets which require logical reasoning. The paper also provides an ablation study and attempts to glance at the inner working of the model by qualitative analysis of attentions. The proposed model appears to work better on long contexts, which is also a plus.

## The bad

It is clear that the paper required multiple revisions in the writing process before submission, as it is rife with many grammatical issues and weird constructions. Even if I choose to ignore that fact, a big issue in the framing of the paper is its complete lack of awareness of relation prediction and link prediction literature. A fact introduced is essentially a triple of source, sink and the relation, which is extensively studied in the Knowledge Base (KB) and Graph Neural Networks community. The related work misses to relate to this large body of work. Preliminaries also miss out on important nomenclature of the logical facts being triples and how it relates to having a knowledge base. In fact, the logic aware pre-training tasks are also variations of popular knowledge base reasoning tasks, such as relation prediction ("Logical Connective Masking" in the paper) and link prediction ("Logical Structure Completion"). The authors should acknowledge this vast area of research, and relate their tasks with the existing literature appropriately.

The proposed pre-training technique appears to work well in downstream tasks, but the paper raises more questions of the entire experimental results. Firstly, it is not clear whether the authors pre-trained the whole model on their pre-training objectives, or just continued pre-training on a publicly available checkpoint. If it is the latter, then the claim that their pre-training objectives are able to inject logical reasoning awareness is significantly weakened, as the downstream performance could be just an artefact of regularization through continual training. The presented results are also incomplete, as no information is provided on whether its the mean or median (typically downstream experiments are reported on a median of five runs, see Liu at al 2019 https://arxiv.org/abs/1907.11692 .) The authors tend to investigate primarily on GLUE benchmark, but it has been shown extensively in the literature that many tasks in this benchmark does not require logical reasoning, as the data consists of annotation artefacts which are leveraged by the model during training. (Gururangan et al 2018 https://arxiv.org/abs/1803.02324, Poliak et al 2019 https://aclanthology.org/S18-2023/, Tsuchiya et al 2018 https://aclanthology.org/L18-1239/, McCoy et al. 2019 https://arxiv.org/abs/1902.01007) Interestingly enough, recent research indicates these tasks might not even require the knowledge of word order, either during fine-tuning (Gupta et al 2021 https://ojs.aaai.org/index.php/AAAI/article/view/17531, Pham et al 2021 https://arxiv.org/abs/2012.15180) or in the pre-trained representation (Sinha et al 2021 https://arxiv.org/abs/2104.06644.) Thus, authors could have instead doubled down on more tasks that require explicit logical reasoning, such as CFQ (https://openreview.net/pdf?id=SygcCnNKwr), CLUTRR (https://arxiv.org/abs/1908.06177), TextWorld (https://arxiv.org/abs/1806.11532), etc.

The analysis presented in the paper (Attention Matrix Heatmap and Case Study) is not convincing and instead raises questions on the motivation and methodology used. Both are cases of extreme cherry picking, which results in a weak argument and does not answer the question of whether the proposed model really imbibes logical reasoning abilities adequately.

## Questions & Comments

### Section 3.1

- Does the fact only represent the two notions "who-did-what-to-whom" and "who-is-what"? There are more categories of relations in knowledge base triples.

### Section 4.2

- "apparently presents in sentences" -> what does this mean?
- The authors had previously defined a fact in Section 3.1 (T={A_1, P, A_2}), yet they do not re-use them in the text ("Argument-Predicate-?" could have been T={A_1, P, ?}). Thus, it does not make sense when new notations are introduced ($m^{a}$ and $m^{p}$)

### Section 4.3

- This section really puzzled me for a while. The authors note "We pre-train our model for 500k steps", and in the next to next sentence they say "Initialized by pre-trained weights of BERT_{base}, we continue training our models for 200k steps." Which one is true? Do the authors just used a pre-trained model (BERT_{base}) and then trained on their logic pre-training objective for 200k steps? If so, then that severely impacts the contribution of the paper as then the benefit of the pre-training tasks are masked by the initialization!

### Table 1, Table 2, Table 5

- It is not clear what these numbers represent - is it mean or median of multiple runs? Or is it a single run on the fine-tuning scores? The authors can consult Liu et al 2019. (RoBERTa) for a good reference on how to report the numbers.

### Section 5.2 Results

- "continual trained for 200K steps for a fair comparison" - this isn't clear. Does the authors mean they trained the baseline BERT_{base} for 200k steps?
- Performing better on GLUE benchmark does not indicate a models better ability for reasoning, as these tasks have been extensively studied in the literature to contain annotation artefacts which the models exploit (Gururangan et al 2018, McCoy et al 2019, etc).
- The claim that the model is robust as it gains consistent improvements on small and large datasets is also not clear to me. Robustness is typically measured by the model's ability to comprehend on different adversarial examples.

### Section 6.2

- Yet another introduction of a new notation, without re-using the ones already defined

### Section 6.3

- I have severe issues with this analysis. First of all, its a cherry-picked result of a single sentence. Secondly, attentions behave differently in different heads of the originally pre-trained BERT model. From the same paper authors cite (Clark et al 2019), the last layers of BERT typically act as a "no-op" operation. It does not make sense to compare the last layer with that of the proposed model to claim the model pays more attention to discourse level information, as due to new training objectives the attention maps might be simply re-arranged in different layers. Attention map comparison of a multi-headed BERT model to display linguistic prowess is also incorrect - there exists other methods to quantify the same, such as the probing literature.

### Section 6.4

- "slope of the dashed line is more gentle" -> how is it quantified? Is it purely based on visual inspection? A metric would be more convincing to quantify the rate of decline.
- Similarly as of Table 1 and 2, the results of Figure 4 cannot be parsed properly as they lack error bars.

### Section 6.5

- The case study is also an instance of cherry-picking. It is good to see the proposed model being more robust to negation and entity change. However, it is hardly convincing as it is a single instance. If the authors want to establish this as a strength of the model they should perform rigorous examination of the entire evaluation set, using these intervention methods.

## Grammatical issues

### Abstract

- "logic reasoning ability" -> "logical reasoning ability"
- "knowledge basis" -> "knowledge bases"
- "logic reasoning" -> "logical reasoning" (lots of instances in the whole paper)
- "We evaluate our model on a broad range of NLP and NLU tasks" - NLU is a subset of NLP, so this statement doesn't make sense. Either "broad range of NLP" tasks or "broad range of NLU tasks"

### Section 2.2

- "multi-granularity pre-training" -> "granular pre-training"

### Section 3.1

- Is "actees" even a word?

### Section 6.2

- "performance is hurt a lot" -> usage of colloquial english is generally discouraged in scientific literature. Consider replacing by "perform is hurt significantly"



**Summary Of The Paper:**

The paper proposes a new pre-training technique to induce a logical prior in the language model representation. Concretely, they propose pre-training on facts, represented as knowledge base triples (source, sink, relation) (knowledge-base completion) and link prediction, alongside traditional masked language modeling objective. Their proposed method achieves some improvement over downstream tasks, including a subset of GLUE benchmark and a couple of relation prediction datasets.



**Summary Of The Review:**

Overall, the paper requires extensive re-writing and better result presentation to be considered for acceptance in the conference. While the paper introduces a nice pre-training technique, it raises way more questions than it answers. Ultimately, the empirical results (and especially the presentation of those results) are not convincing enough to support the claim that the proposed pre-training techniques at all induces logical reasoning capabilities to BERT model. At this stage, unfortunately I cannot recommend acceptance.

---

### Official Review · Reviewer_cXhg · 2021-11-08

**Correctness:** 3
**Technical Novelty And Significance:** 4
**Empirical Novelty And Significance:** 3
**Recommendation:** 5
**Confidence:** 4

**Main Review:**

Strengths
- Good and clear ablations to verify each pre-training objective, along with an assessment of longer term dependencies by context length
- Approach and findings are important for the community in that this work uses no external knowledge bases for incorporating logical reasoning, positioned well relative to previous work

Weaknesses
- The emphasis on self-supervision could be stronger in the paper to make clear that it is a self-supervised approach to logical reasoning
- The submission has areas where the technical explanation could be clearer (see questions below), and even the naming convention of the proposed pre-training objectives is not consistent
- Several methodological choices were not justified (see questions below)
- The attention matrix heatmap analysis is not convincing due to certain methodological choices

Questions
- How exactly do you convert the dependency parse to the triplet of {$A_1, P, A_2$}? Please illustrate with an example
- Why do you choose to model the logical graph as edges defined over relations between argument-predicates and between coreferents only? Are other relationships possible? Why did you not choose them?
- In equation 3, there is no indication that the final training objective is weighted? Was a weighted sum used? What weights did you try and why?
- In 4.3, it is unclear which model is pre-trained for 500k steps and which is fine-tuned from BERT for 200k steps.
- If the baseline model is fine-tuned for 200k steps, why is that a sufficiently fair comparison with your pre-trained model with 500k steps?
- In table 2: why are PROPHET's results clearly better than BERT large on LogiQA but perform worse / about the same on ReClor?
- Why do you only choose the first head of the last attention layer for the attention matrix? What do you see for the other heads or other layers? I'm not convinced by this methodological choice

Presentation concerns
- Spelling of stabilized in Figure 1
- If you specify what V is in 3.2, it is also good to specify what E is
- The names used in Figure 2 for pre-training objective should correspond with those in 4.2
- In 5.2, advise to keep Table 2 results separate from the list of Table 1 results
- Suggest you show the BERT baseline in Table 4 for comparison with the ablations
- In Table 6, are the figures from vanilla BERT or Prophet?
- In Figure 5, typo "nentailment"


**Summary Of The Paper:**

The goal of the paper is to incorporate logical relations into pre-training of language models to solve the reliance of existing reasoning-enabled language models on external knowledge bases. This is done in a self-supervised way - facts (tuple of 2 arguments and a predicate) are parsed using dependency parsing, and then a logical graph is created to denote relationships between coreferents, and between predicates and arguments. Three pre-training objectives are presented over the facts and logical graph: logical connective masking, logical structure completion and logical path prediction.

The author's claimed contributions are the following:
- 3 new pre-training objectives: logical connective masking, logical structure completion, logical path prediction
- The model Prophet which "achieves significant improvement over various logic reasoning involved NLP and NLU downstream tasks"
- An analysis that verifies how Prophet is using the context for logical reasoning

**Summary Of The Review:**

Overall, I think this paper is very marginally below the acceptance threshold. I like the self-supervised approach to logical reasoning in pre-training. My major concern is the clarity of the paper, which hinders some explanation of methodological choices and proper understanding of the technical approach. I am willing to move this paper above the acceptance threshold if the authors can address my questions appropriately.

---

### Decision · Program_Chairs · 2022-01-20

**Decision:**

Reject

**Comment:**

This paper proposes a pre-training technique for improving the logical abilities of pre-trained language models.
Reviewers point to many issues with clarity and experimental evaluation. No response was given by authors.